



# On the Limits of Köhler Activation Theory: How do Collision and Coalescence Affect the Activation of Aerosols?

Fabian Hoffmann

Institute of Meteorology and Climatology, Leibniz Universität Hannover, Hannover, Germany.

*Correspondence to:* F. Hoffmann (hoffmann@muk.uni-hannover.de)

**Abstract.** Activation is necessary to form a cloud droplet from an aerosol, and it occurs as soon as a wetted aerosol grows
beyond its critical radius. Traditional Köhler theory assumes that this growth is driven by the diffusion of water vapor. However,
if the wetted aerosols are large enough, the coalescence of two or more particles is an additional process for accumulating
sufficient water for activation. This transition from diffusional to collectional growth marks the limit of traditional Köhler
theory and it is studied using a Lagrangian cloud model in which aerosols and cloud droplets are represented by individually
simulated particles within large-eddy simulations of shallow cumuli. It is shown that the activation of aerosols larger than
$0.1\,\mu$m in dry radius can be affected by collision and coalescence, and its contribution increases with a power-law relation
toward larger radii and becomes the only process for the activation of aerosols larger than $0.4 - 0.8\,\mu$m depending on aerosol
concentration. Due to the natural scarcity of the affected aerosols, the amount of aerosols that are activated by collection is
small with a maximum of 1 in $10\,000$ activations. The fraction increases as the aerosol concentration increases, but decreases
again as the number of aerosols becomes too high and the particles too small to cause collections. Moreover, activation by
collection is found to affect primarily aerosols that have been entrained above the cloud base.

## 1  Introduction

Activation is necessary for the formation of droplets from aerosols. Accordingly, activation controls the number and size of
cloud droplets and hence so-called aerosol-cloud interactions, e.g., cloud albedo (Twomey, 1974) or cloud lifetime (Albrecht,
1989). In contrast to cloud droplets, which behave like bulk water, the understanding of unactivated aerosols and their activa-
tion depends fundamentally on the aerosol's physicochemical properties, which cause the so-called solute and curvature effects
(Köhler, 1936). These effects enable, on the one hand, the stable existence of haze particles (also termed wetted aerosols) in
subsaturated environments and inhibit, on the other hand, diffusional growth if the supersaturation does not exceed a certain
threshold. This so-called critical supersaturation is associated with a critical radius, to which a wetted aerosol must grow to be
considered as activated. Small aerosols activate almost immediately when the supersaturation exceeds the critical supersatura-
tion, as it is assumed in many parameterizations of the activation process (e.g., Twomey, 1959). For larger aerosols, however,
the critical radius becomes so large that the time needed for activation can be substantially increased (or even prevented un-
der certain conditions) due to the kinetically limited transport of water vapor to the particle's surface (Chuang et al., 1997).
Therefore, Köhler activation theory is usually considered a weak concept for these particles. But where are the limits of Köhler





activation theory located? An upper limit of the applicability of Köhler activation theory can be identified by the switch from
predominantly diffusional to collectional (collision followed by coalescence) mass growth if the involved particles become
large enough. Indeed, inactivated aerosols triggering collisions is closely related to the impact of giant and ultra-giant aerosols
(dry radius $> 1\,\mu$m) on clouds, which are able to initiate precipitation due to their large wet radii ($> 20\,\mu$m) (e.g., Johnson,
1982). Recent studies indicate that collection might even affect smaller particles: by considering the effects of turbulence, the
collection kernel for the interaction of small particles can be significantly increased (e.g., Devenish et al., 2012). Accordingly,
the main questions of this study are: Where are the limits of traditional Köhler theory? At which aerosol size will collection
dominate the activation process? And how much does collectional activation contribute to the activation of aerosols? To an-
swer these questions, theoretical arguments and large-eddy simulations (LES) with particle-based cloud physics are applied.
Particle-based cloud physics, so-called Lagrangian cloud models (LCMs), are especially suitable for this study because they
explicitly resolve the activation process and do not rely on a parameterization of it (e.g., Andrejczuk et al., 2008; Hoffmann
et al., 2015; Hoffmann, 2016). Therefore, the results will give insights on the physical processes usually not covered (or missed)
by those activation parameterizations typically implemented in other cloud models.
This paper is designed as follows. The subsequent Section 2 will illuminate how collections can cause (or even inhibit)
activation by simple theoretical arguments. In Section 3, the LES-LCM simulation setup is introduced. Results will be presented
in the Sections 4 and 5, where the former section exemplifies the applied methodology used to untangle diffusional from
collectional activation and the latter section presents the results from a shallow cumulus test case. The study is summarized and
discussed in Section 6. Appendix A introduces the governing equations of the applied LCM and necessary extensions carried
out for this study.

## 45  2   Theoretical considerations

In this section, the general effects of coalescence on the activation of aerosols will be addressed. To simplify the argumentation
in this part of the study, it is assumed that collections take place regardless of the physics that enable or inhibit them in reality.
We consider one particle which grows by coalescing with other particles. Accordingly, the particle's water mass after $n$
collections is given by
$$m_n = m_0 + \sum_{i=1}^{n} m_i = m_0 + n \cdot \langle m \rangle, \tag{1}$$
where $m_0$ terms the particle's initial water mass and $m_i$ ($i > 0$) the mass of water added by each collection. The second equals
sign introduces the assumption of a monodisperse ensemble of collected particles.
Based on Köhler theory, it can be shown that the critical radius for activation is given by
$$r_{\mathrm{crit}} = \sqrt{3\frac{b \cdot m_{\mathrm{s}}}{A}}, \tag{2}$$
where $m_{\mathrm{s}}$ is the dry aerosol mass. Curvature effects are considered by $A = 2\sigma/(\rho_{\mathrm{l}} R_{\mathrm{v}} T)$, depending on the surface tension of
water $\sigma$, mass density of water $\rho_{\mathrm{l}}$, specific gas constant of water vapor $R_{\mathrm{v}}$, and temperature $T$. The physicochemical aerosol





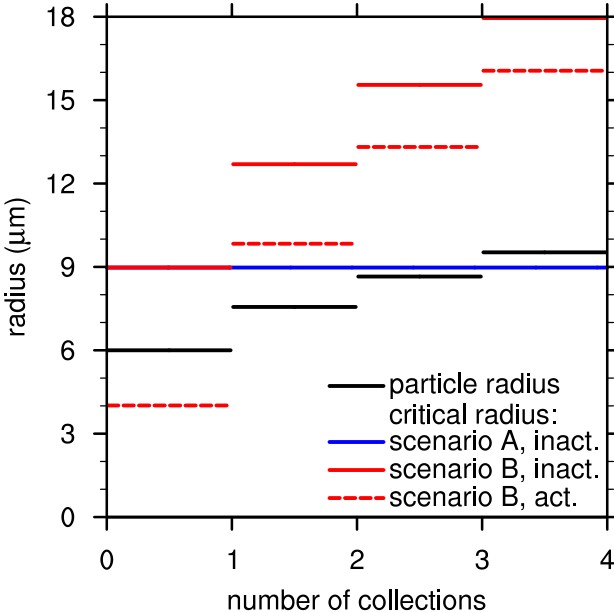

**Figure 1.** Change of particle radius (black line) and critical radius (colored lines) as a function of the number of collections for the growth scenarios A (negligible increase of aerosol mass, blue line) and B (aerosol mass increases proportional to the number of collections, red lines) as well as initially inactivated (continuous lines) and activated particles (dashed line). The initial wet particle radius and the wet radii of the collected particles are assumed to be $6\,\mu$m. The initial dry aerosol mass (sodium chloride) is $2.2 \times 10^{-16}\,$kg ($0.29\,\mu$m dry radius) (continuous lines) and $4.4 \times 10^{-17}\,$kg ($0.17\,\mu$m dry radius) (dashed line). For scenario B, the collected particles contain $2.2 \times 10^{-16}\,$kg dry aerosol mass ($0.29\,\mu$m dry radius).

properties responsible for the solute effect are represented by $b = 3\nu_s\rho_s\mu_l/(4\pi\rho_l\mu_s)$, with the van't Hoff factor $\nu_s$, the mass
density of the aerosol $\rho_s$, and the molecular masses of water $\mu_l$ and aerosol $\mu_s$, respectively. Accordingly, the critical mass for
activation after $n$ collections yields
$$m_{\mathrm{crit},n} = \frac{4}{3}\pi\rho_l \cdot r_{\mathrm{crit},n}^3 = \frac{4}{3}\pi\rho_l \cdot \left[ 3\frac{b}{A} \cdot \left( m_{s,0} + \sum_{i=1}^{n} m_{s,i} \right) \right]^{3/2}, \tag{3}$$

where $m_{s,0}$ terms the initial aerosol mass and $m_{s,i}$ $(i > 0)$ the aerosol mass added by each collection. Approximating the
summation in (3) demands further assumptions on the distribution of aerosol mass within the particle spectrum. Two scenarios
are defined. Scenario A: the collected particles contain a negligible amount of aerosols. Accordingly, the aerosol mass does
not change ($\sum_{i=1}^{n} m_{s,i} = 0$). Scenario B: each particle contains the same mass of aerosol. Correspondingly, the aerosol mass
increases proportionally to the number of collections ($\sum_{i=1}^{n} m_{s,i} = n \cdot \langle m_s \rangle$).
In Fig. 1, the evolving particle radius and critical radius are displayed as a function of the number of collections (details on
the particle properties are given in the figure's caption). The simultaneous examination of particle radius and critical radius
reveals if a particle is activated (particle radius larger than critical radius) or deactivated (particle radius smaller than critical





radius). For scenario A, the initially inactivated particle (black line) grows faster than the critical radius (blue line), and the
aerosol activates after 3 collections. For scenario B, an initially inactivated particle (continuous red line) and an initially
activated particle (dashed red line) are examined. Since the critical radius for activation increases faster than the particle radius,
activation is inhibited or the deactivation of previously activated particle is caused.
These considerations suggest that only the collection of particles with a large amount of water and a comparably small
amount of aerosol mass (i.e., highly dilute solution droplets) might lead to activation (as shown in scenario A). This, however,
indicates that the collected particles are probably activated already. Therefore, the process of collectional activation will not in-
crease the total number of activated aerosols since one ore more already activated aerosols need to be collected (or annihilated)
in the process of collectional activation. By contrast, the collection of particles with a comparably large amount of aerosol
(i.e., less dilute solutions, as shown in scenario B) might inhibit activation since the increase of the critical radius exceeds the
increase of the wet radius.
The following part of the study is investigating how coalescence is able to cause aerosol activation in shallow cumulus clouds
using a detailed cloud model considering diffusional growth as well as detailed physics of collision and coalescence.

## 82   3   Simulation setup

The following results are derived from LES simulations applying an LCM for representing cloud microphysics. The LCM is
based on a recently developed approach which simulates individual particles that represent an ensemble of identical particles
and maintains, as an inherent part of this approach, the identity of droplets and their aerosols throughout the simulation (An-
drejczuk et al., 2008; Shima et al., 2009; Sölch and Kärcher, 2010; Riechelmann et al., 2012; Naumann and Seifert, 2015). A
summary of the governing equations and the extensions carried out for this study to treat aerosol mass change during collision
and coalescence is given in the Appendix A. The underlying dynamics model, the LES model PALM (Maronga et al., 2015),
solves the non-hydrostatic incompressible Boussinesq-approximated Navier-Stokes equations, and prognostic equations for
water vapor mixing ratio, potential temperature, and subgrid-scale turbulence kinetic energy. For scalars, a monotonic advec-
tion scheme (Chlond, 1994) is applied to avoid spurious oscillations at the cloud edge (e.g., Grabowski and Smolarkiewicz,

92 1990).

The initial profiles and other forcings of the simulation follow the shallow trade wind cumuli intercomparison case by
Siebesma et al. (2003), which itself is based on the measurement campaign BOMEX (Holland and Rasmusson, 1973). A
cyclic model domain of $3.2 \times 3.2 \times 3.2 \, \mathrm{km}^3$ is simulated. (In comparison to Siebesma et al. (2003), the horizontal extent has
been halved in each direction due to limited computational resources.) The grid spacing is $20 \, \mathrm{m}$ isotropically. Depending on
the prescribed aerosol concentration, a constant time step of $\Delta t = 0.2 - 0.5 \, \mathrm{s}$ had to be used for the correct representation of
condensation and evaporation, but it is also applied to all other processes. The first $1.5$ hours of simulated time are regarded as
model spin-up; only the following four hours are analyzed.
The simulated particles, called super-droplets following the terminology of Shima et al. (2009), are released at the beginning
of the simulation, and are randomly distributed within the model domain up to a height of $2800 \, \mathrm{m}$. The average distance between




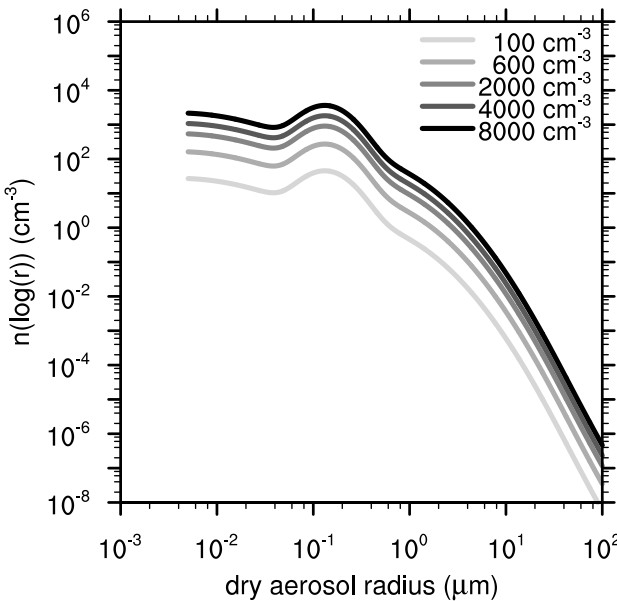

**Figure 2.** The number density distribution of dry aerosol radii for different aerosol concentrations (line brightness).

the super-droplets is $4.3\,\mathrm{m}$, yielding a total number of about $360 \times 10^6$ simulated particles and about $100$ super-droplets per
grid box. Initial weighting factors, i.e., the number of real particles represented by each super-droplet, are $8 \times 10^9$, $48 \times 10^9$,
$160 \times 10^9$, $320 \times 10^9$, and $640 \times 10^9$, representing aerosol concentrations of $100$, $600$, $2000$, $4000$, and $8000\,\mathrm{cm}^{-3}$, respectively.
These result in average droplet concentrations of $48$, $220$, $550$, $750$, and $1000\,\mathrm{cm}^{-3}$, respectively.
The dry aerosol radius is assigned to each super-droplet using a random generator which obeys a typical maritime aerosol
distribution represented by the sum of three lognormal distributions (Jaenicke, 1993) (Fig. 2). However, only aerosols larger
than $0.005\,\mu\mathrm{m}$ are initialized since smaller aerosols do not activate in the current setup. The different aerosol concentrations
are created by scaling the weighting factor of each simulated particle to attain the desired concentration. The aerosols are
assumed to consist of sodium chloride (NaCl, mass density $\rho_\mathrm{s} = 2165\,\mathrm{kg\,m}^{-3}$, van't Hoff factor $\nu_\mathrm{s} = 2$, molecular weight $\mu_\mathrm{s} =$
$58.44\,\mathrm{g\,mol}^{-1}$). The initial wet radius of each super-droplet is set to its approximate equilibrium radius depending on aerosol
mass and ambient supersaturation (Eq. (14) in Khvorostyanov and Curry, 2007). The applied collection kernel includes effects
of turbulence, which have been shown to increase the collection probability of small particles significantly (e.g., Devenish
et al., 2012). See Appendix A for more details.

## 4 Methodology

In this section, the applied methodology for untangling the contributions of diffusion and collection to the activation of aerosols
is introduced. An aerosol becomes activated when it grows beyond its critical radius ($r > r_\mathrm{crit}$). This process can be driven by the
diffusion of water vapor or by accumulating liquid water due to collection or by a combination of both. To enable unhindered





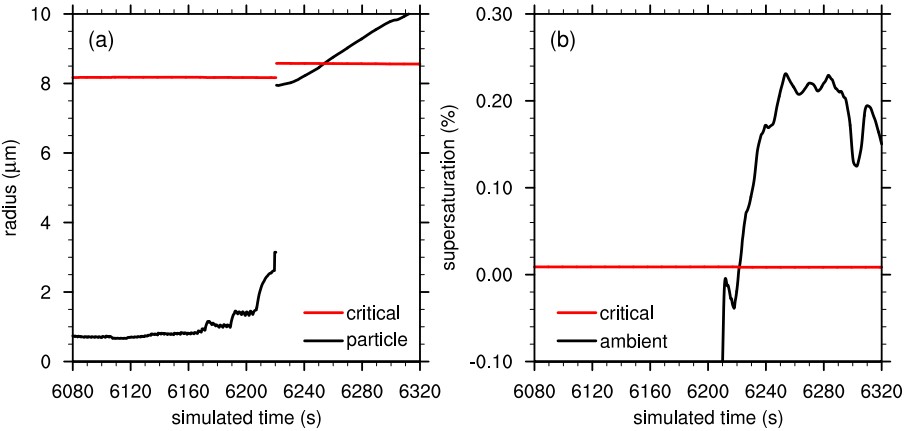

**Figure 3.** Time series of a particle which is activated by collection. Panel (a) shows its radius (black) and critical radius (red) and panel (b) depicts the ambient supersaturation experienced by that particle (black) and its critical supersaturation (red).

diffusional growth after activation, the activated particle is required to be located in a volume of air which exceeds the critical
supersaturation at the moment of activation ($S > S_{\text{crit}}$ at $r = r_{\text{crit}}$). This is always fulfilled in the case of diffusional growth, but
it is checked additionally in the case of collectional activation to ensure equivalence of collectional and diffusional activation.
To decide if an activation is primarily driven by diffusion or collection, all simulated particles have been tracked throughout
the simulation and their mass growth has been integrated from their minimum mass before activation, $\min(m)$, to the critical
activation mass, $m_{\text{crit}}$:
$$\Delta m|_{\text{diff}} = \int_{\min(m)}^{m_{\text{crit}}} \mathrm{d}m|_{\text{diff}},\qquad\qquad(4)$$

$$\Delta m|_{\text{coll}} = \int_{\min(m)}^{m_{\text{crit}}} \mathrm{d}m|_{\text{coll}},\qquad\qquad(5)$$

where $\mathrm{d}m|_{\text{diff}}$ and $\mathrm{d}m|_{\text{coll}}$ are directly derived from the LCM's model equations (A2) and (A5) – (A6), respectively. Note the
following procedures for determining $\min(m)$, $\Delta m|_{\text{diff}}$, and $\Delta m|_{\text{coll}}$ during the simulation: (i) If a particle shrinks below
$\min(m)$ before activation, $\Delta m|_{\text{diff}}$ and $\Delta m|_{\text{coll}}$ are set to zero and are re-calculated starting from this new minimum mass.
(ii) If a particle becomes deactivated, i.e., evaporates smaller than its critical radius after being activated, the current mass is
considered the new $\min(m)$ and $\Delta m|_{\text{diff}}$ and $\Delta m|_{\text{coll}}$ are set to zero. (iii) If a collection does not result in an activation and
the particle evaporates back to its equilibrium radius afterwards, $\Delta m|_{\text{diff}}$ will be negative and $\Delta m|_{\text{coll}}$ positive. To avoid the
potentially incorrect classification of a following activation, $\Delta m|_{\text{diff}}$ and $\Delta m|_{\text{coll}}$ are set to zero if $\Delta m|_{\text{diff}}$ becomes negative
and the current mass is considered as $\min(m)$.
The following two processes are considered a collectional activation if the collectional mass growth exceeds the diffusional
($\mathrm{d}m|_{\text{coll}} > \mathrm{d}m|_{\text{diff}}$): first, the coalescence of two inactivated aerosols resulting directly or after some diffusional growth in an





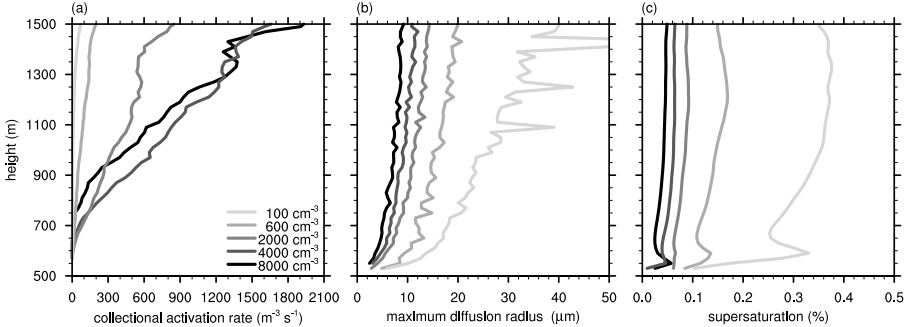

**Figure 4.** Vertical profiles of the collectional activation rate (a), the maximum diffusion radius (b), and the supersaturation (c) for the analyzed aerosol concentrations (line brightness).

activation; second, the coalescence of an inactivated aerosol with an activated aerosol resulting in an inactivated aerosol, which
activates after some diffusional growth. If the latter process results directly in an activated aerosol, this collection is only
considered a collectional activation if the wet radius of initially activated particle is smaller than the critical radius of the
newly formed activated particle. The latter restriction ensures that the coalescence of both particles is necessary to aggregate
the required amount of water for activation and excludes scavenging by large activated particles collecting smaller ones while
precipitating. Note that only collections of the first type are able to increase the number of activated aerosols, while the second
type might have no or a negative impact on the total number of activated aerosols as discussed in Section 2.

144        To exemplify this methodology, Fig. 3 shows, for an aerosol selected from the LCM simulations discussed below, the time

series of its radius and critical radius (panel a) and the ambient supersaturation and critical supersaturation (panel b). Note that
this aerosol is actually one super-droplet, representing a larger ensemble of identical aerosols, which is, however, interpreted as
one aerosol here. The initial dry radius of the aerosol is $0.27\,\mu$m. On its way to activation, the particle experiences diffusional
growth, which can be easily identified by the continuous change of radius. One collection event, characterized by a distinct
increase in radius, is visible at $6220\,$s simulated time. At this point in time, the inactivated aerosol (wet radius $3.1\,\mu$m) coalesces
with an activated particle (wet radius $7.8\,\mu$m, aerosol dry radius $0.13\,\mu$m), but the product of coalescence (wet radius $7.9\,\mu$m,
aerosol dry radius $0.28\,\mu$m) remains inactivated. Due to the increased amount of aerosol mass, the critical radius (and to a lesser
extent the critical supersaturation) increases (decreases) after the coalescence. Afterwards, the particle grows by diffusion and
exceeds the critical radius at $6253\,$s simulated time, which can be identified as the time of activation. All in all, this activation
is considered a collectional activation since $\mathrm{d}m|_{\mathrm{coll}} = 1.9 \times 10^{-12}\,\mathrm{kg} > \mathrm{d}m|_{\mathrm{diff}} = 6.2 \times 10^{-13}\,\mathrm{kg}$.

## 5    Results

The last section showed that collection can contribute significantly to the mass growth leading to the activation of a single
aerosol. But how does collection contribute to the activation of aerosols in general? Figure 4 shows the vertical profiles of
(a) the collectional activation rate, i.e., the number of aerosols activated by collection per unit volume and unit time, (b) the



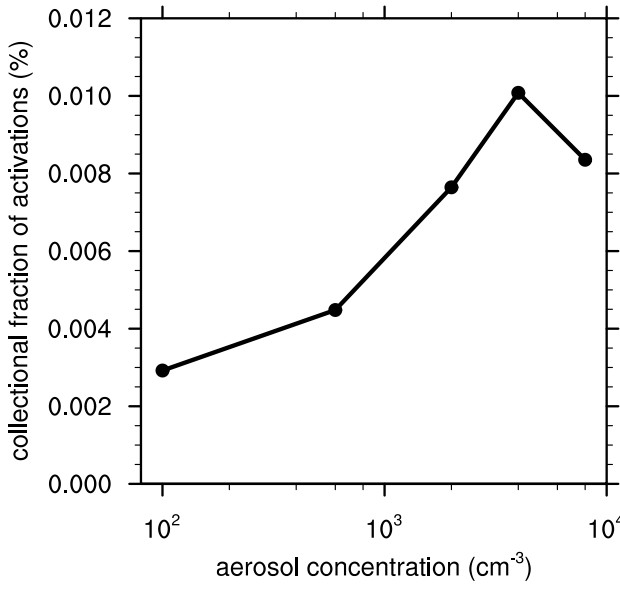

**Figure 5.** The collectional fraction of all activations as a function of the aerosol concentration.

maximum diffusion radius, i.e., the maximum critical radius of aerosols exclusively activated by diffusion at a certain height,
and (c) the supersaturation. Profiles (a) and (c) are conditionally averaged over all supersaturated grid cells. Only data of the
last 4 simulated hours is considered. Values above the average cloud top height (at $1500\,\mathrm{m}$) are not displayed due to insufficient
statistics.
The maximum diffusion radius (Fig. 4 b) increases (neglecting outliers) monotonically with height reaching maxima between
$40\,\mu\mathrm{m}$ and $9\,\mu\mathrm{m}$ for aerosol concentrations of $100\,\mathrm{cm}^{-3}$ to $8000\,\mathrm{cm}^{-3}$, respectively. The supersaturation (Fig. 4 c) exhibits
a distinct peak at the cloud base and relaxes toward its equilibrium value determined by the number of activated aerosols
and vertical velocity above (e.g., Rogers and Yau, 1989, Chap. 7). Due to the larger number of water vapor absorbers, the
supersaturation as well as the maximum diffusion radius are generally smaller in the more aerosol-laden simulations.
The collectional activation rate (Fig. 4 a) increases almost linearly with height. This increase can be related to the longer
lasting diffusional growth resulting in potentially larger particles at higher levels, which increases the collection kernel and
therefore the collection probability. The slope is larger in aerosol-laden environments, where more aerosols are available
for activation. Additionally, the height above cloud base where the collectional activation starts increases with the aerosol
concentration since the average particle radius is too small to enable collisions at lower levels. Accordingly, the collectional
activation rate in the $8000\,\mathrm{cm}^{-3}$ simulation exhibits smaller to similar values than in the $4000\,\mathrm{cm}^{-3}$ simulation although the
slope in the $8000\,\mathrm{cm}^{-3}$ simulation is larger. Note that the general shape of the collectional activation rate differs significantly
from the typical profile of diffusional activation, which exhibits as a distinct peak at cloud base where the majority of aerosols
activates by diffusion (not shown, see, e.g., Slawinska et al., 2012; Hoffmann et al., 2015).





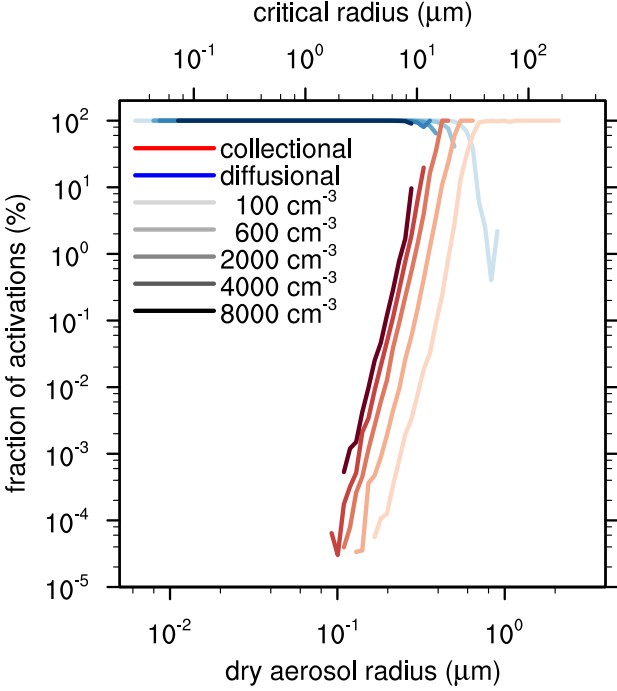

**Figure 6.** The collectional (red lines) and diffusional (blue lines) fraction of activations as a function of the dry aerosol radius (lower abscissa) and critical radius (at cloud base temperature of $294.5\,\mathrm{K}$, upper abscissa) for the analyzed aerosol concentrations (line brightness).

Generally, the contribution of collectional activation to the number of activated aerosols is significantly smaller than the
contribution of diffusional activation (Fig. 5): only 1 activation in $10\,000$ to $35\,000$ is caused by collection, with a greater
contribution of collectional activation in moderately aerosol-laden environments up to $4000\,\mathrm{cm}^{-3}$. As it will be outlined below,
this increase can be attributed to a shift of collectional activation to smaller, but more numerous aerosols. For $8000\,\mathrm{cm}^{-3}$,
however, the fraction decreases again since the particles are too small to trigger a larger amount of collisions.
Figure 6 shows the collectional and diffusional fraction of activations as a function of the dry aerosol radius on the lower
abscissa and the corresponding critical radius (calculated for the cloud base temperature of approximately $294.5\,\mathrm{K}$) on the
upper abscissa. As expected, diffusional activation is the dominant process for small aerosols (dry radius $< 0.1\,\mu\mathrm{m}$) as long
as the dry aerosol radius is not too small and the corresponding critical supersaturation not too high to inhibit activation.
Accordingly, the left boundary of diffusional activation is shifted toward larger radii as the maximum supersaturations decrease
in more aerosol-laden environments (see Fig. 4 c). For aerosols larger than $0.1\,\mu\mathrm{m}$, collectional activation becomes increasingly
important affecting aerosols in the range of $0.16 - 2.5\,\mu\mathrm{m}$, $0.13 - 0.65\,\mu\mathrm{m}$, $0.11 - 0.46\,\mu\mathrm{m}$, $0.092 - 0.33\,\mu\mathrm{m}$, $0.11 - 0.28\,\mu\mathrm{m}$
for aerosol concentrations of $100$, $600$, $2000$, $4000$, and $8000\,\mathrm{cm}^{-3}$, respectively. Larger aerosols do not activate at all.
The collectional fraction of activations increases following a power-law relation toward larger radii, reflecting the higher
collision probability of larger particles. The collectional fraction reaches up to $100\,\%$ for the $100$, $600$, and $2000\,\mathrm{cm}^{-3}$ simula-





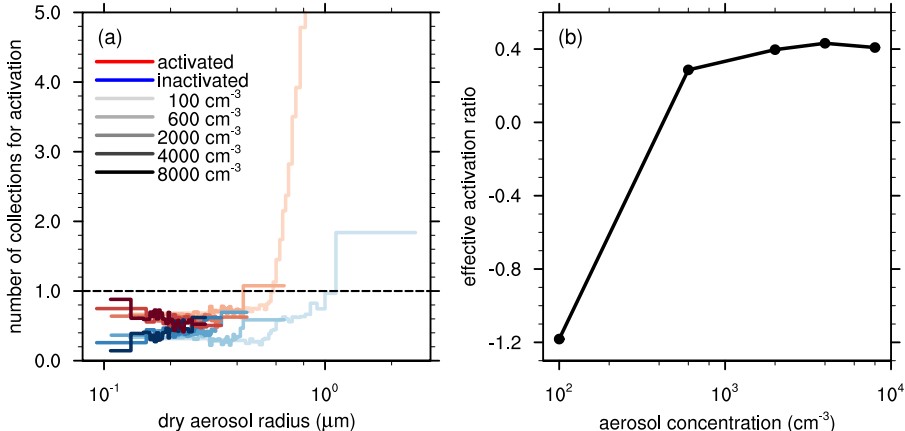

**Figure 7.** Panel (a) displays the number of collected activated (red lines) and inactivated aerosols (blue lines) necessary to cause collectional activation as a function of the dry aerosol radius for the analyzed aerosol concentrations (brightness). The data has been binned; each bin contains at least $3\%$ of all registered collectional activations. Panel (b) shows the effective activation ratio (i.e., the net increase in the number of newly activated aerosols per collectional activation) as a function of aerosol concentration.

tions at about $0.83$, $0.54$, and $0.42\,\mu$m dry aerosol radius, respectively, indicating a significant effect of collectional activation on this part of the aerosol spectrum. For higher aerosol concentrations, collectional activation does not dominate, but still contributes noteworthy with fractions up to $20\%$ and $10\%$ for aerosol concentrations of $4000$ and $8000\,\text{cm}^{-3}$, respectively. The dry aerosol radius at which activation reaches $100\%$ can be clearly assigned to the maximum radii that can be produced by diffusion. To create any larger particles, existing particles need to be merged. Accordingly, to activate aerosols with a larger critical radius, collection must be inherently involved. For the $100\,\text{cm}^{-3}$ simulation, the largest radii produced by diffusion are about $40\,\mu$m (neglecting the outliers in Fig. 4 b), corresponding to a dry aerosol radius of $0.76\,\mu$m, which is close to the dry aerosols exhibiting a $100\%$ collectional fraction of activations. A similar agreement can be found for the simulations initialized with aerosol concentrations of $600$ and $2000\,\text{cm}^{-3}$.

In general, the range of aerosols affected by collectional activation shifts toward smaller radii as the aerosols concentration increases. This is primarily a result of the decreasing maximum radii that can be reached by diffusion alone (Fig. 4 b). Additionally, the supersaturation decreases too (Fig. 4 c), which decelerates diffusional activation and therefore favors collectional activation. Since small aerosols are significantly more abundant than larger ones (Fig. 2), the number of aerosols that are potentially activated by collection increases as a result of this shift, resulting in the larger collectional fraction of all activations shown in Fig. 5.

In Section 2, it has been argued that the collection of particles with a large fraction of liquid water (and accordingly less aerosol) are more beneficial to collectional activation than particles with a large amount of aerosol mass. Figure 7 a displays the average number of collisions that take place during a collectional activation, separated into collected activated and collected inactivated particles. Accordingly, their sum yields the total number of collected particles necessary for a collectional activation.

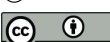



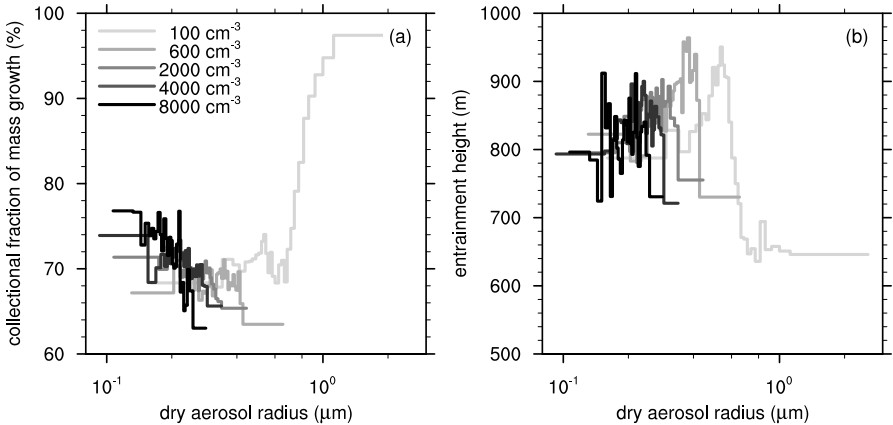

**Figure 8.** Collectional fraction of (a) the mass growth leading to collectional activation, and (b) the average entrainment height as a function of the dry aerosol radius for the analyzed aerosol concentrations (brightness). The data has been binned; each bin contains at least $3\%$ of all registered collectional activations.

For dry aerosol radii up to $0.3 - 0.5\,\mu$m (depending on aerosol concentration), only one collection (activated plus inactivated)
is necessary to cause activation, while for larger aerosols more collections are needed. For the aerosols activated by only one
collision, about $40\%$ of all events involve two inactivated aerosols and $60\%$ an inactivated as well as one activated aerosol,
indicating the beneficial effect of highly dilute solution droplets to collectional activation as discussed above.
Accordingly, a substantial number of activated aerosols are annihilated during collectional activation. To quantify the influ-
ence of collectional activation on the number of activated aerosols, the *effective activation ratio* is defined: the net increase in
the number of newly activated aerosols per collectional activation. Figure 7 b displays the effective activation ratio calculated
from all registered collectional activations. For an aerosol concentration of $100\,\text{cm}^{-3}$, where a large portion of aerosols needs
multiple collections for activations, the effective activation ratio is $-1.2$, i.e., more activated aerosols are annihilated than pro-
duced. But already for an aerosol concentration of $600\,\text{cm}^{-3}$ and more, the effective activation ratio becomes positive and is
approximately constant at $0.4$, indicating that per collectional activation an average number of $0.4$ new activated aerosols are
produced. This ratio has to be considered in the interpretation of Fig. 5, indicating that the net effect of collectional activation
is actually smaller (or even negative).
Although activation is dominated by collectional mass growth for larger aerosols, the growth by diffusion is still essential to
create sufficiently large particles to trigger collisions. Figure 8 a depicts the collectional fraction of mass growth needed to grow
beyond the critical mass for activation (for aerosols activated by collection). Note that the diffusional fraction of mass growth
is the remaining fraction. For the smallest affected aerosols ($\sim 0.1\,\mu$m), the collectional fraction of mass growth is about $75\%$
and decreases slightly to $65\%$ for aerosols of $\sim 0.4\,\mu$m, indicating that a large contribution of diffusional growth is necessary
to produce sufficient large particles that are able to collide. The slight decrease toward larger radii is in agreement with the
decrease in the number of activated aerosols collected during the activation process (Fig. 7 a): collection is only possible for



the smallest aerosols if they encounter a substantially larger activated particle, which results in a larger collectional fraction
of mass growth and a larger number of collected activated aerosols. For aerosols larger than $1\,\mu$m, the collectional fraction
increases rapidly to $97\,\%$, which can be attributed to the large critical radii which can be only exceeded by the collection of
multiple droplets.
Figure 8 b displays the mean entrainment height of the particles involved in each collectional activation. Despite the largest
particles ($> 0.6\,\mu$m) in the most pristine case ($100\,\mathrm{cm}^{-3}$), all collectional activations involve particles that have entered the
cloud well above the cloud base, which is located at $500 - 600\,$m. Accordingly, these particles miss the typical supersaturation
maximum located at cloud base (see Fig. 4 c), where a majority of these aerosols normally activates. Indeed, entrainment above
cloud base is generally favorable for collectional activation since these aerosols are mixed into an environment where larger
particles exist, triggering collisions among them more easily. For aerosols larger than $0.6\,\mu$m, the average entrainment height
is located closer to the cloud base. Since multiple collections are necessary for their activation (see Fig. 7 a), the lower average
entrainment height is more representative for the average entrainment height of all particles inside the cloud, which is the cloud
base (e.g., Hoffmann et al., 2015).

## 244  6   Summary and discussion

The influence of collision and coalescence on the activation of aerosols has been studied using theoretical arguments and large-
eddy simulations (LES) with a coupled Lagrangian cloud model (LCM). The presented theory has shown that an unactivated
aerosol can be activated by the collection of particles with a comparably small amount of aerosol mass (i.e., particles consisting
almost entirely of water), while the collection of large amounts of additional aerosol mass inhibits activation or even causes the
deactivation of previously activated aerosols. The LCM simulations of shallow trade wind cumuli indicated that collectional
activation becomes possible for aerosols larger than approximately $0.1\,\mu$m in dry radius, and its contribution increases with a
power-law relation toward larger aerosols. In pristine conditions, collection is the only process for the activation of aerosols
larger than $0.83\,\mu$m in dry radius at an aerosol concentration of $100\,\mathrm{cm}^{-3}$. This boundary is shifted to smaller radii in more
polluted environments (down to $0.42\,\mu$m at $2000\,\mathrm{cm}^{-3}$). The highest contribution of collectional activation to the total number
of activated aerosols is found at an aerosol concentration of $4000\,\mathrm{cm}^{-3}$, where 1 in $10\,000$ activations is caused by collec-
tion. If the aerosol concentration becomes higher and hence the particles too small, collectional activation is inhibited and its
contribution decreases again. Collectional activation frequently involves the collection of already activated aerosols reducing
the net increase of newly activated aerosols per collectional activation to $0.4$, while the remainder ($0.6$ activated aerosols) is
annihilated during the activation process. Moreover, collectional activation affects predominantly particles that have been en-
trained above cloud base, i.e., activates aerosols that have not been able to activate by diffusion at cloud base, where the largest
supersaturations occur. Finally, it has been shown that the collectional activation rate increases almost linear with height, while
the slope and the height, from which collectional activation starts, increase with the aerosol concentration.
In conclusion, this study revealed collision and coalescence as an additional process for the activation of aerosols. This
process is not covered by commonly applied activation parameterizations (e.g., Twomey, 1959). But does this matter? First



of all, with a maximum of 1 in 10 000 activations, collectional activation can be safely neglected. But one can also argue
that collectional activation is already (but implicitly) covered by standard cloud models: Activation parameterizations usually
activate aerosols as soon as the critical supersaturation is exceeded, i.e., they neglect kinetic effects inhibiting the immediate
activation of large aerosols, which need a certain time to grow beyond their critical radius. As pointed out by Chuang et al.
(1997), this might overestimate the number of activated aerosols (or cloud droplets) since a certain fraction of the larger
aerosols is falsely treated as activated (or as cloud droplets). However, following the argumentation of Nenes et al. (2001),
these particles might act, due to their large wet radii, as regular cloud droplets although they are not formally activated, and the
estimated droplet number concentration is not influenced by this shortcoming of the activation parameterization. And indeed,
this study showed that a certain fraction of these formally inactivated particles are able to collide and coalesce, i.e., act as
regular cloud droplets. Similarly, in standard cloud models, these falsely activated cloud droplets will experience the model's
representation of collision and coalescence that might ultimately result in an implicit realization of collectional activation.
Accordingly, collectional activation is not of particular importance for determining the number of cloud droplets, but it
indicates clearly the limits of Köhler activation theory. Without ambiguity, diffusion-based Köhler theory is only applicable
to aerosols smaller than $0.1\,\mu$m in dry radius, while an increasing fraction of aerosols activates by collection at larger radii.
Ultimately, the activation of aerosols larger than about $1.0\,\mu$m is entirely caused by collection (if it takes place at all). Therefore,
the range between approximately $0.1\,\mu$m and $1.0\,\mu$m should be considered as a transition zone between (i) typical aerosols that
need to experience sufficiently strong supersaturations to grow beyond the critical radius and (ii) so-called giant and ultra-
giant aerosols with sufficiently large wet radii to act like cloud droplets by triggering collision and coalescence without being
formally activated (e.g., Johnson, 1982).
Finally, potential sources of uncertainty within this study shall be mentioned. First, the accuracy of the applied collection
kernel is limited. The widely-used collision efficiencies of Hall (1980) for small particles ($\lesssim 20\,\mu$m) are slightly higher than
other estimates (e.g., Böhm, 1992). An effect of this uncertainty is the collectional activation of aerosols that are too small
to collide physically. Accordingly, collectional activation shall affect slightly larger radii than evaluated here. Further note
that additional simulations neglecting turbulence effects on the collection kernel (not shown) have exhibited a similar spectral
distribution of collectional activation, but indicated a smaller contribution to the total number of activated aerosols. Second,
the initialized aerosol distribution is always maritime, i.e., it includes a large fraction of large aerosols which are not part of
continental air masses (e.g., Jaenicke, 1993) but are primarily affected by collectional activation as shown here. Accordingly,
the collectional fraction of activations might be lower in environments which exhibit a smaller fraction of aerosols in the
affected size range. Third, not all aerosols consist of (highly hygroscopic) sodium chloride although the size range affected by
collectional activation is usually assumed to consists of sea salt (Jaenicke, 1993). Aerosols with a lower hygroscopicity would
exhibit a smaller solution effect which is equivalent to a smaller dry radius of the sodium chloride aerosols examined here,
i.e., the wet radius of these aerosols would be smaller and they would less likely cause collisions. Again, the range of aerosols
affected by collectional activation would be shifted to larger radii.





## Appendix A: The Lagrangian cloud model

In this section, the basic framework of the Lagrangian cloud model (LCM) applied in this study as well as the extensions made to treat aerosol mass during collision and coalescence are described. One can refer to Riechelmann et al. (2012) for the original description, Hoffmann et al. (2015) for the consideration of aerosols during diffusional growth, and Hoffmann et al. (2017, in review) for the most recent description of the LCM. This LCM, as all other available particle-based cloud physical models (Andrejczuk et al., 2008; Shima et al., 2009; Sölch and Kärcher, 2010; Naumann and Seifert, 2015), are based on the so-called *super-droplet* approach in which each simulated particle represents an ensemble of identical, real particles, growing continuously from an aerosol to a cloud droplet. The number of particles within this ensemble, the so-called *weighting factor*, is a unique feature of each particle, which is considered for a physical appropriate representation of cloud microphysics within the super-droplet approach.

The transport of a simulated particle is described by

$$\frac{\mathrm{d}X_i}{\mathrm{d}t} = u_i + \widetilde{u}_i - \delta_{i3} w_\mathrm{s}, \tag{A1}$$

where $X_i$ is the particle location and $u_i$ is the LES resolved-scale velocity at the particle location determined from interpolating linearly between the 8 adjacent grid points of the LES. A turbulent velocity component $\widetilde{u}_i$ is computed from a stochastic model based on the LES sub-grid scale turbulence kinetic energy (Sölch and Kärcher, 2010). The sedimentation velocity $w_\mathrm{s}$ is given by an empirical relationship (Rogers et al., 1993). Equation (A1) is solved using a first-order Euler method.

As described in Hoffmann et al. (2015), the diffusional growth of each simulated particle is calculated from

$$r\frac{\mathrm{d}r}{\mathrm{d}t} = \frac{S - A/r + b \cdot m_\mathrm{s}/r^3}{F_\mathrm{k} + F_\mathrm{D}} \cdot f(r, w_\mathrm{s}), \tag{A2}$$

where $r$ is the particle's radius and $S$ terms the supersaturation within the grid box, in which the particle is located. Curvature and solution effects are considered by the the terms $-A/r$ and $b \cdot m_\mathrm{s}/r^3$, respectively. The factor $f$ parameterizes the so-called ventilation effect (Rogers and Yau, 1989). The coefficients $F_\mathrm{k} = (L_\mathrm{v}/(R_\mathrm{v}T) - 1) \cdot L_\mathrm{v}\rho_\mathrm{l}/(Tk)$ and $F_\mathrm{D} = \rho_\mathrm{l}R_\mathrm{v}T/(D_\mathrm{v}e_\mathrm{s})$ represent the effects of thermal conduction and diffusion of water vapor between the particle and the surrounding air, respectively. Here, $k$ is the coefficient of thermal conductivity in air, $D_\mathrm{v}$ is the molecular diffusivity of water vapor in air, $L_\mathrm{v}$ is the latent heat of vaporization, and $e_\mathrm{s}$ is the saturation vapor pressure. Equation (A2) is solved using a fourth-order Rosenbrock method.

Collision and coalescence are calculated from a statistical approach in which collections are calculated from the particle size distribution resulting from all super-droplets currently located within a grid box (Riechelmann et al., 2012). These interactions affect the weighting factor $A_n$ (i.e., the number of all particles represented by one super-droplet), the total water mass of a super-droplet $M_n = A_n \cdot m_n$ (where $m_n$ is the mass of one particle represented by super-droplet $n$), and also the dry aerosol mass $M_{\mathrm{s},n} = A_n \cdot m_{\mathrm{s},n}$ (where $m_{\mathrm{s},n}$ is the dry aerosol mass of one particle represented by super-droplet $n$), which has been introduced for this study. The algorithm follows the *all-or-nothing* principle, which has been rigorously evaluated by Unterstrasser et al. (2016, in review) and has been recently implemented into this LCM by Hoffmann et al. (2017, in review).

It is assumed that the super-droplet with the smaller weighting factor (index $n$) collects $A_n$ particles from the super-droplet with the larger weighting factor (index $m$), with commensurate changes in $M_m$, $M_n$, $M_{\mathrm{s},m}$, and $M_{\mathrm{s},n}$. Since the weighting





factor of the collecting super-droplet $n$ does not change during this process, its wet radius
$$r_n = \left( \frac{M_n}{\frac{4}{3}\pi \rho_l A_n} \right)^{1/3} \tag{A3}$$
and the dry aerosol radius
$$r_{s,n} = \left( \frac{M_{s,n}}{\frac{4}{3}\pi \rho_s A_n} \right)^{1/3} \tag{A4}$$
increase. Additionally, same-size collections of the particles belonging to the same super-droplet are considered. These inter-
actions do not change $M_n$ and $M_{s,n}$, but they decrease $A_n$ and accordingly increase $r_n$ and $r_{s,n}$.
These two processes yield in the following description for the temporal change of $A_n$ (assuming that the simulated particles
are sorted such that $A_n > A_{n+1}$):
$$\frac{\mathrm{d}A_n}{\mathrm{d}t}\delta t = -\frac{1}{2}(A_n - 1)P_{nn} - \sum_{m=n+1}^{N_p} A_m P_{mn}. \tag{A5}$$
The first term on the right-hand-side denotes the loss of $A_n$ due to same-size collections; the second term the loss of $A_n$ due
to collisions with particles of a smaller weighting factor. The total water mass and the total aerosol mass of a super-droplet
change according to
$$\frac{\mathrm{d}M_n}{\mathrm{d}t}\delta t = \sum_{m=1}^{n-1} A_n m_m P_{nm} - \sum_{m=n+1}^{N_p} A_m m_n P_{mn}, \tag{A6}$$
and
$$\frac{\mathrm{d}M_{s,n}}{\mathrm{d}t}\delta t = \sum_{m=1}^{n-1} A_n m_{s,m} P_{nm} - \sum_{m=n+1}^{N_p} A_m m_{s,n} P_{mn}, \tag{A7}$$
respectively. In both equations, the first term on the right-hand-side denotes the increase of $M_n$ or $M_{s,n}$ by the collection of
water or dry aerosol mass from super-droplets with a larger weighting factor, while the second term describes the loss of these
quantities to super-droplets with a smaller weighting factor. The function $P_{mn}$ controls if a collection takes place:
$$P_{mn} := P(\varphi_{mn}) = \begin{cases} 0 & \text{for } \varphi_{mn} \le \xi, \\ 1 & \text{for } \varphi_{mn} > \xi, \end{cases} \tag{A8}$$
where $\xi$ is a random number uniformly chosen from the interval $[0,1]$ and
$$\varphi_{mn} = K(r_m, r_n, \epsilon) A_n \delta t / \Delta V \tag{A9}$$
is the probability that a particle with the radius $r_m$ collects one of $A_n$ particles with the radius $r_n$ within a volume $\Delta V$ during
the (collection) time step $\delta t$. The collection kernel $K$ is calculated from the traditional collision efficiencies as given by Hall
(1980), and includes turbulence effects by an enhancement factor for the collision efficiencies by Wang and Grabowski (2009)



and a parameterization of particle relative velocities and changes in the particle radial distribution based on Ayala et al. (2008).
These turbulence effects on $K$ are steered by the dissipation rate $\epsilon$ calculated by the LES subgrid-scale model. The equations
(A5) – (A7) are solved using a first-order Euler method.
*Acknowledgements.* The author thanks Siegfried Raasch and Katrin Scharf (both of the Leibniz Universität Hannover) for their helpful
comments on the manuscript. This work has been funded by the German Research Foundation (DFG) under grant RA 617/27-1. Simulations
have been carried out on the Cray XC-40 systems of the North-German Supercomputing Alliance (HLRN). The applied LES/LCM model is
freely available (revision 1954, http://palm.muk.uni-hannover.de/trac/browser/?rev=1954). Additional software developed for the LES/LCM
model as well as the analysis is available on request.



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
