# Peer review of "On the Limits of Köhler Activation Theory: How do Collision and Coalescence Affect the Activation of Aerosols?"

_Atmospheric Chemistry and Physics, 2017_

## Referee Comment (RC1) · Anonymous Referee #1 · 30 Mar 2017

The manuscript discusses the process of activation of cloud droplets on big aerosol particles. It checks for what aerosol size range the process of activation of cloud droplets can be explained by collisions between aerosol particles. It also checks the importance of the process of activation via collection compared to activation via diffusion of water vapor. The study is done using an LES setup combined with Lagrangian (i.e. particle tracking) representation of cloud droplets are represented using the Lagrangian approach, which allows to numerically resolve the activation process.

As shown in the manuscript, the studied process of activation by collection is very rare and affects mostly big aerosol particles entrained above the cloud base. As discussed

in the summary of the manuscript, the studied process can be safely neglected, or even more, it might already be implicitly covered in some of the activation parametrization schemes. The presented study is therefore more theoretical and shows, in my understanding, in what aerosol size range the term *activation* as understood by the Köhler theory has any meaning.

The manuscript is well written and my further comments are both few and minor.

**General comments**

The manuscript defines three scenarios of activation of an aerosol particle by collision (lines 135-143):

- 1. coalescence of two inactivated aerosol particles resulting directly or after some diffusional growth in activated particle,
- 2. coalescence of an inactivated aerosol particle and activated aerosol particle that leads to an inactivated particle that activates due to diffusion,
- 3. coalescence of an inactivated aerosol particle and activated aerosol particle that leads to an activated particle. This scenario is considered an activation via collection only when the critical radius of the created particle is bigger than the initial wet radius of the colliding activated aerosol.

The first scenario is straightforward, but in my opinion the second and the third scenario deserve more explanation why they are considered an activation via collection. Indeed, from the point of view of the colliding inactivated aerosol particle, it can be said that the activated aerosol particle with which it collided got annihilated and in turn the aerosol in question got activated after some additional diffusional growth. However, from the point of view of the colliding activated particle it can be said that the activated aerosol particle scavenged the inactivated particle and thanks to diffusion of water vapor remained activated (i.e. the activated particle remains activated and the inactivated particle is annihilated).

In general, counting and labeling activation events that happen due to collision is more difficult because there are two initial particles and one resulting activated aerosol particle, whereas the traditional Köhler theory activation results in one-to-one correspondence between an activated aerosol particle and the created cloud droplet. Could you clarify which colliding particles are considered activated and which annihilated?

Could you consider adding some sketch or maybe a plot using Köhler curves that exemplifies how the considered scenarios work? It could help to clarify which particles are labeled as annihilated, activated and inactivated and to showcase the typical dry and wet radius sizes of the particles colliding in all scenarios.

**Specific comments**

- **line 26**: As discussed in the Summary when referring to the work by Nenes et al. 2001, it is not necessary for a cloud droplet to become formally activated (i.e. reach its critical radius as defined by the Köhler theory) in order to grow in the cloudy environment and behave similar to the formally activated droplets. Could you consider adding such comment also in the introduction?
- **line 32**: I think the question this article addresses is about "limits of traditional Köhler **activation** theory". As discussed in the Summary and in the referred work of Chuang et al. 1997 and Nenes et al. 2001, the Köhler theory can be used to calculate the equilibrium saturation for big aerosol particles. The problem is that

СЗ

the big aerosol particles will not reach their equilibrium in the necessary time and therefore will not become formally activated.

- **line 103**: Does it mean that the weighting factors for all super-droplets are constant? Does it affect the representation of collisions (compared to the tests presented in Unterstrasser et al. 2016)?
- **Figure 4**: I think the panels should be bigger (at least as big as those in Fig. 3). What is causing the spikes for maximum diffusion radius for the simulation with the lowest aerosol concentration? For convenience, would you consider adding a panel that shows the diffusional activation rate calculated basing on the simulations discussed here?
- Figure 7a and lines 207-214: Figure 7a is difficult for me to read and understand. First, the lines are plotted on top of each other making it difficult to see the behavior of each simulation. Second, the description of what is on the axes and what is actually plotted is unclear to me. For example, in the simulation with 4000 aerosols in cm-3 for dry radius of 0.1  $\mu$ m there are 0.2 collisions with inactivated aerosol particles and 0.8 collisions with activated aerosol particles to activate the aerosol particle. In the same time in the description it is stated that only one collision is needed to cause activation and that the collision occurs between an activated and inactivated particle. Could you clarify, or maybe provide some example? Third, are all aerosol particles counted twice in this plot? Once as the aerosol particle that is going to be activated (i.e. the location on the x-axis) and once as colliding particles (i.e. the different lines shown on the plot)?
- **line 258-259**: "collectional activation affects predominantly particles that have been entrained above cloud base, i.e., activates aerosols that have not been able to activate by diffusion at cloud base (...)" Does this sentence mean that the aerosols in question were not activated at cloud base because they were never

at the cloud base? If yes, then I think saying that those aerosols have not been able to activate at cloud base is misleading, because they were never there.

- line 353: Could you clarify what values of dissipation rate were used for the collision efficiency from the Wang and Grabowski 2009 paper? The efficiencies in this paper are provided for two dissipation rates (either 100 cm2/s3 or 400 cm2/s3). Was the closer one chosen? Or was a constant dissipation rate assumed when choosing the collision efficiency?
- **line 356**: Would you consider Brownian motion of aerosol particles as another possibility for activation due to collisions? Would a collision kernel representing both Brownian motion of aerosol particles and turbulence effects be an interesting extension of this study?

**Technical corrections**

- line 23 and 39: I would not use the word *even* when describing opposite behavior(?).
- **line 195**: When saying activation you mean collectional activation? Maybe it should be explicitly stated?
- **line 326**: I think that the paper by Shima et al. 2009 should be referred here again when introducing the "all-or-nothing" representation of collisions for the Lagrangian microphysics.

---

## Referee Comment (RC2) · S Shima (Referee) · 14 Apr 2017

I would like to recommend this paper to be published but after major revisions.

This is an interesting paper introducing a new mechanism of cloud droplet activation named "collectional activation". The author investigated its contribution theoretically, then numerically using an LCM. Note also that this analysis could only be possible if using an LCM. One of the conclusion is that the impact is small because it seldom occurs compared to conventional "diffusional activation", but I think the community still needs to be aware of such possibility.

However, there exist at least one major issue in this manuscript. Unfortunately, the

determination criterion of "collectional activation" the author introduced is not appropriate. Please see the attached note "Possible_collectional_activation_scenario.pdf". You can see that r>r_crit is not a rigorous criterion to determine "collectional activation". I strongly suggest the author to examine all the materials minutely, keeping the above fact in mind, I am still not fully sure how big the revision could be, but because all the analyses are based on the above criterion, this correction could affect the paper substantially, though it probably do not change the main conclusion significantly.

Please also see other major/minor comments annotated in the attached pdf.

Please also note the supplement to this comment:
http://www.atmos-chem-phys-discuss.net/acp-2017-134/acp-2017-134-RC2-supplement.zip

―――――――――――――――――――

---

## Author Comment (AC1)

**Response to Anonymous Referee #1**

The manuscript discusses the process of activation of cloud droplets on big aerosol particles. It checks for what aerosol size range the process of activation of cloud droplets can be explained by collisions between aerosol particles. It also checks the importance of the process of activation via collection compared to activation via diffusion of water vapor. The study is done using an LES setup combined with Lagrangian (i.e. particle tracking) representation of cloud microphysics. In the discussed simulations both aerosol particles and cloud droplets are represented using the Lagrangian approach, which allows to numerically resolve the activation process.

As shown in the manuscript, the studied process of activation by collection is very rare and affects mostly big aerosol particles entrained above the cloud base. As discussed in the summary of the manuscript, the studied process can be safely neglected, or even more, it might already be implicitly covered in some of the activation parametrization schemes. The presented study is therefore more theoretical and shows, in my understanding, in what aerosol size range the term *activation* as understood by the Köhler theory has any meaning.

The manuscript is well written and my further comments are both few and minor.

*Thank you very much for your comments which helped to clarify the manuscript.*

**General comments**

The manuscript defines three scenarios of activation of an aerosol particle by collision (lines 135-143):

1. coalescence of two inactivated aerosol particles resulting directly or after some diffusional growth in activated particle,

2. coalescence of an inactivated aerosol particle and activated aerosol particle that leads to an inactivated particle that activates due to diffusion,

3. coalescence of an inactivated aerosol particle and activated aerosol particle that leads to an activated particle. This scenario is considered an activation via collection only when the critical radius of the created particle is bigger than the initial wet radius of the colliding activated aerosol.

The first scenario is straightforward, but in my opinion the second and the third scenario deserve more explanation why they are considered an activation via collection. Indeed, from the point of view of the colliding inactivated aerosol particle, it can be said that the activated aerosol particle with which it collided got annihilated and in turn the aerosol in question got activated after some additional diffusional growth.

However, from the point of view of the colliding activated particle it can be said that the activated aerosol particle scavenged the inactivated particle and thanks to diffusion of water vapor remained activated (i.e. the activated particle remains activated and the inactivated particle is annihilated).

In general, counting and labeling activation events that happen due to collision is more difficult because there are two initial particles and one resulting activated aerosol particle, whereas the traditional Köhler theory activation results in one-to-one correspondence between an activated aerosol particle and the created cloud droplet. Could you clarify which colliding particles are considered activated and which annihilated?

Could you consider adding some sketch or maybe a plot using Köhler curves that exemplifies how the considered scenarios work? It could help to clarify which particles are labeled as annihilated, activated and inactivated and to showcase the typical dry and wet radius sizes of the particles colliding in all scenarios.

*The identification of collectional mass growth is based on the comparison of the collectional mass growth $\Delta m|_{coll}$ to the diffusional $\Delta m|_{diff}$. The scenarios exemplify how this $\Delta m|_{coll}$ is able to exceed $\Delta m|_{diff}$. Accordingly, they are scenarios defined, they result from the collections I observed. And indeed, the first scenario is straight forward, but the other scenario might also lead to $\Delta m|_{coll} > \Delta m|_{diff}$, and need to be considered. I added a sketch to the manuscript (Fig. 3) which illustrates each scenario. The sketch displays the critical radius (red) as well as the wet radius (blue) of each particle during the process of collectional activation. A more in depth discussion of the relevant processes have been added to the text (line 149-166):*

*"To identify a collectional activation, the integrated collectional mass growth $\Delta m|_{coll}$ is compared to the diffusional $\Delta m|_{diff}$ in the moment the particle grows beyond its critical radius. If the former exceeds the latter, $\Delta m|_{coll} > \Delta m|_{diff}$, this activation is considered as collectional. There are various microphysical interactions resulting in $\Delta m|_{coll} > \Delta m|_{diff}$, and its basic types are illustrated in Fig. 3. Note that also a combination or a repetition of these types is possible, i.e., multiple subsequent collections. In a collectional activation of type (i), the water mass growth by collection dominates, i.e., the coalescence of two previously inactivated aerosols A and B results directly or after some diffusional growth in an activated particle C. In a collectional activations of type (ii), the critical radius increases faster than wet radius, i.e., the coalescence of an already activated particle A with another activated or an inactivated particle B results in inactivated particle C, which activates after some diffusional growth.  If the resulting particle is directly activated, this process is only considered a collectional activation if the largest wet radius of the two coalescing particles A and B is smaller than the critical radius of the newly produced particle C:*
$$max(r_A, r_B) < r_{crit,C}.$$
*This ensures that the combined water of particles A and B is necessary to activate particle C. If this is not the case, i.e., the water of particle A or B is able to activate particle C on its own, the latter process is considered a regular collection of cloud droplets or as scavenging and neglected in the following analysis. Moreover, the coalescence of two activated*

*particles resulting in a collectional activation is mathematically possible but not found to play a role in the analyzed simulations. Note that only collectional activations of the first type are able to increase the number of activated aerosols, while the second type might have no or a negative impact on the total number of activated aerosols since the coalescence of at least one activated particle results in one activated particle."*

[Figure]

**Specific comments**

• **line 26**: As discussed in the Summary when referring to the work by Nenes et al. 2001, it is not necessary for a cloud droplet to become formally activated (i.e. reach its critical radius as defined by the Köhler theory) in order to grow in the cloudy environment and behave similar to the formally activated droplets. Could you consider adding such comment also in the introduction?

*Yes (line 25 - 27): "Due to their large size, however, these particles may behave like regular cloud droplets inside the environment of a cloud although they are not formally activated (Nenes et al., 2001). Accordingly, Köhler activation theory is usually considered a weak concept for these particles."*

• **line 32**: I think the question this article addresses is about "limits of traditional Köhler

**activation** theory". As discussed in the Summary and in the referred work of Chuang et al. 1997 and Nenes et al. 2001, the Köhler theory can be used to calculate the equilibrium saturation for big aerosol particles. The problem is that the big aerosol particles will not reach their equilibrium in the necessary time and therefore will not become formally activated.

*You are perfectly right. I added the word "activation" to clarify this (line 34).*

• **line 103**: Does it mean that the weighting factors for all super-droplets are constant? Does it affect the representation of collisions (compared to the tests presented in Unterstrasser et al. 2016)?

*Initially, the weighting factors are the same, which might impede collections in a zero-dimensional setup as tested in Unterstrasser et al. (2017). Accordingly, the results should be considered as a lower estimate of the impact of collectional activation. However, as super-droplets experience collections, their weighting factor reduces resulting in a wide range of different weighting factors during the simulation. As discussed in Unterstrasser et al. (2017), this might facilitate collisions if more than one grid box is simulated, i.e., if super-droplets are allowed to interact with another ensemble of droplets when they move from one grid box to the next. The following addition has been made (line 314 - 315): "Additionally, the collection algorithm itself might underestimate collisions due to the initial distribution of weighting factors (Unterstrasser et al., 2017), and the determined influence of collectional activation should be considered as a lower estimate."*

• **Figure 4**: I think the panels should be bigger (at least as big as those in Fig. 3). What is causing the spikes for maximum diffusion radius for the simulation with the lowest aerosol concentration? For convenience, would you consider adding a panel that shows the diffusional activation rate calculated basing on the simulations discussed here?

*The size of the panels has been increased (Fig. 5). A panel of the diffusional activation has been added (Fig. 5 d), which caused some subsequent changes in the text (line 198 - 200).*

*Thank you for the hint regarding the spikes in the 100 cm$^{-3}$ simulation. They result from the recirculation of large particles (see Naumann and Seifert, 2016, doi: 10.1002/2016MS000631), which have grown by collection inside the cloud, then detrained from the cloud, evaporated smaller than their critical radius outside the cloud (i.e., deactivated), entrained into the cloud again, where they grew larger than the critical radius by diffusion (i.e., activated by diffusion). Since the algorithm for distinguishing between diffusional and collectional activation only considered the growth between deactivation and activation, they have been spuriously considered as diffusional activations. In total, only 2 x 10$^{-4}$ % of all diffusional activation have been affected by this process. I was able to remove these false diffusional activations from the analysis of the 100 cm$^{-3}$ simulation. No influence of recirculations has been found for simulations with a*

*higher aerosol concentration. The new profile for the 100 cm$^{-3}$ simulation has been added to Fig. 5. The conclusions did not change.*

• **Figure 7a and lines 207-214**: Figure 7a is difficult for me to read and understand. First, the lines are plotted on top of each other making it difficult to see the behavior of each simulation. Second, the description of what is on the axes and what is actually plotted is unclear to me. For example, in the simulation with 4000 aerosols in cm$^{-3}$ for dry radius of 0.1 µm there are 0.2 collisions with inactivated aerosol particles and 0.8 collisions with activated aerosol particles to activate the aerosol particle. In the same time in the description it is stated that only one collision is needed to cause activation and that the collision occurs between an activated and inactivated particle. Could you clarify, or maybe provide some example? Third, are all aerosol particles counted twice in this plot? – Once as the aerosol particle that is going to be activated (i.e. the location on the x-axis) and once as colliding particles (i.e. the different lines shown on the plot)?

*The whole figure has been changed to clarify the manuscript. Figure 8 shows the average number of collections necessary for activation irrespective of the number of collected activated or inactivated particles. All necessary information on how many activated aerosols have been involved in a collectional activation was already contained in Fig. 9 (the former Fig. 7b).*

• **line 258-259**: "collectional activation affects predominantly particles that have been entrained above cloud base, i.e., activates aerosols that have not been able to activate by diffusion at cloud base (...)" Does this sentence mean that the aerosols in question were not activated at cloud base because they were never at the cloud base? If yes, then I think saying that those aerosols have not been able to activate at cloud base is misleading, because they were never there.

*You are right. The sentence has been clarified to: "Moreover, collectional activation affects predominantly particles that have been entrained above cloud base, i.e., above the region of the cloud where the highest supersaturations occur. Accordingly, these particles experience systematically lower supersaturations which prevents diffusional activation." (line 280 - 282)*

• **line 353**: Could you clarify what values of dissipation rate were used for the collision efficiency from the Wang and Grabowski 2009 paper? The efficiencies in this paper are provided for two dissipation rates (either 100 cm2/s3 or 400 cm2/s3). Was the closer one chosen? Or was a constant dissipation rate assumed when choosing the collision efficiency?

*The kinetic energy has been determined in the sub-grid scale model of the LES and the efficiencies of Wang and Grabowski (2009) have been interpolated to that value (using the given data for 100 cm$^2$ s$^{-3}$, 400 cm$^2$ s$^{-3}$, as well as unity for a zero dissipation rate). This has been clarified to: "These turbulence effects are steered by the kinetic energy dissipation rate $\epsilon$ calculated in the LES subgrid-scale model (Riechelmann et al., 2012). The*

*parameterizations by Ayala et al. (2008) are a direct function of ε, while the tabulated values of the enhancement factor for the collision efficiency by Wang and Grabowski (2009) are interpolated to the present value of ε." (line 384 - 386)*

• **line 356**: Would you consider Brownian motion of aerosol particles as another possibility for activation due to collisions? Would a collision kernel representing both Brownian motion of aerosol particles and turbulence effects be an interesting extension of this study?

*Indeed, a collision kernel with Brownian motions and turbulence would be an interesting extension of this study. Especially for very small collected particles, the consideration of additional processes affecting the collection process might result into a larger fraction of collected particles (e.g., Ardon-Dryer et al., 2015, doi: 10.5194/acp-15-9159-2015). However, I would expect that Brownian motions would rather have no impact on collisional activation by facilitating the collection of aerosols with a negligible amount of liquid water but a comparably large fraction of aerosol mass. This would result in a faster increase of the critical radius than the wet radius and therefore inhibit collisional activation (as discussed in Section 2 of the manuscript). I added a short discussion to Section 6 (line 309 - 312): "Moreover, the collection kernel might not incorporate all processes relevant for collections among aerosols and droplets. For instance, Brownian diffusion might increase the collection of smaller particles (e.g., Ardon-Dryer et al., 2015) but might not lead to collectional activation since it will add predominantly aerosol mass and only a small amount of water (cf. Section 2)"*

**Technical corrections**

• **line 23 and 39**: I would not use the word *even* when describing opposite behavior(?).

   *Ok. The word "even" is not necessary there.*

• **line 195**: When saying activation you mean collectional activation? Maybe it should be explicitly stated?

   *Done.*

• **line 326**: I think that the paper by Shima et al. 2009 should be referred here again when introducing the "all-or-nothing" representation of collisions for the Lagrangian microphysics.

   *Good point. Done.*

---

## Author Comment (AC2)

**Response to Shin-ichiro Shima**

I would like to recommend this paper to be published but after major revisions.

This is an interesting paper introducing a new mechanism of cloud droplet activation named "collectional activation". The author investigated its contribution theoretically, then numerically using an LCM. Note also that this analysis could only be possible if using an LCM. One of the conclusion is that the impact is small because it seldom occurs compared to conventional "diffusional activation", but I think the community still needs to be aware of such possibility.

However, there exist at least one major issue in this manuscript. Unfortunately, the determination criterion of "collectional activation" the author introduced is not appropriate. Please see the attached note "Possible_collectional_activation_scenario.pdf". You can see that r>r_crit is not a rigorous criterion to determine "collectional activation". I strongly suggest the author to examine all the materials minutely, keeping the above fact in mind, I am still not fully sure how big the revision could be, but because all the analyses are based on the above criterion, this correction could affect the paper substantially, though it probably do not change the main conclusion significantly.

Please also see other major/minor comments annotated in the attached pdf.

*I am very thankful for the reviewer's comments which helped to clarify the paper in various aspects. However, I do not agree with his major comment on the appropriateness of the applied criterion for the detection of collectional activations, which will be outlined in this general response. More detailed answers will follow below.*

*The reviewer argues that the applied criterion to determine if an aerosol is activated or not, i.e., to distinguish between aerosols and cloud droplets, by comparing their radius against their respective critical radius ($r$ vs. $r_{crit}$), is not adequate. In the present manuscript, I consider a particle as activated if it has grown beyond its critical radius ($r > r_{crit}$), a criterion which has been used and applied by various authors before (e.g., Rogers and Yau, 1989; Chuang et al., 1997; Khain et al., 2000; Boucher 2015; Hoffmann et al., 2015). Additionally, I request that the supersaturation enables further diffusional growth in the moment of activation to establish equivalence of diffusional and collisional activation (see line 119 - 123). Accordingly, the reviewer's collectional activation scenarios (ii) to (iv) are already considered in this study, which has been clarified and explained in more detail in the revised version of the manuscript (line 119 - 135 and comment 8 below). I only disagree with the reviewer's scenario (i). The reviewer argues that all particles which experience a supersaturation that exceeds the critical supersaturation ($S > S_{crit}$) should be considered as activated irrespective of their radius.*

*Of course, a supersaturation which exceeds the critical supersaturation ($S > S_{crit}$) will result in a radius which exceeds the critical radius ($r > r_{crit}$) at some point in time. And indeed, if the temporal dimension of particle growth and hence activation is neglected, both criterions are identical (see lines 20 - 27). But the time necessary for activation increases significantly for larger aerosols due to the kinetically limited transport of water molecules to the particle (Chuang et al. 1997; Hoffmann 2016). And if the supersaturation varies, as it*

*is the case in a real cloud due to entrainment/turbulence or simply due to the cloud's limited lifetime, the considered particle might not grow beyond its critical radius although the critical supersaturation has been exceeded for a certain period of time. Accordingly, the criterion of r > $r_{crit}$ is essential to decide if an activation has been completed or not.*

*Moreover, the critical supersaturations of the aerosols affected by collectional activation are so low that they are easily exceeded anywhere inside the cloud (cf. Fig. 5b). For the smallest aerosols affected by collectional activation (0.1 µm dry radius), the critical supersaturation is 0.03 % and decreases significantly for larger ones (e.g., 0.005 % for a radius of 0.4 µm, i.e., where the collectional fraction of activations becomes significant). The following figure shows the average supersaturation at the moment of collectional activation. Accordingly, the critical supersaturation is not restricting activation; it is exceeded several times by the supersaturations found in the simulated clouds.*

[Figure]

*Accordingly, the reviewer's criterion to consider all aerosols with S > $S_{crit}$ as activated makes no sense for the analysis carried out in this study. It would probably consider all aerosols larger than 0.1 µm as activated. And we would have no information if these aerosols succeed to grow beyond the critical radius for activation. (Which is probably not the case due to the kinetically limited transport of water vapor to the particle (e.g., Chuang et al., 1997; Nenes et al., 2001; Hoffmann et al. 2015).) Anyhow, the reviewer's questions shows perfectly the problems associated with Köhler activation theory at these large aerosol radii: It is simply not valid anymore. The critical supersaturation is easily exceeded, but the growth beyond the critical radius can be impeded by the naturally occurring variations of the supersaturation.*

*Further reviewer comments (I copied them in a chronological order from the reviewer's PDF annotations):*

1. Major request.
This is not true for "collectional activation". Modify it appropriately.
*See main response above.*

2. Major request.
The discussion here is interesting and helpful to understand "collectional activation". However, r>r_crit is not a rigorous criterion for "collectional activation". Consider how to revise or justify the analysis.
*See main response above.*

3. Minor request.
To avoid confusion, you should explicitly mention that condensation/evaporation process is ignored in the theoretical analysis in this section.
*Good point: "Moreover, all other microphysical processes, specifically diffusional growth, are neglected." (line 50)*

4. Minor suggestion
To avoid confusion, you should clearly mention that those two red lines represent the critical radii, not the particle radius.
*Good point: "For scenario B, an initially inactivated particle and an initially activated particle are examined (the critical radii are displayed in red by a continuous or dashed line, respectively)." (line 73 - 74)*

5. Major question
Isn't this too big for calculating collision coalescence? Maybe it is okay for your method but have you checked the sensitivity to dt?
*I didn't check the sensitivity to dt in this study, but a general study on the sensitivity of the collection algorithm to dt can be found in Unterstrasser et al. (2017). For a timestep of 1.0 s the results are reasonable. Accordingly, they should also be reasonable for a timestep of less or equal to 0.5 s. A reference to the study of Unterstrasser et al. (2017) is already given in line 356.*

6. Major request
Please make it clear how you decide the initial dry aerosol radius. Uniform random sampling in log(dry_r) space? or any other?
*Yes, as already stated two sentences above: "The dry aerosol radius is assigned to each super-droplet using a random generator which obeys a typical maritime aerosol distribution represented by the sum of three lognormal distributions (Jaenicke, 1993) (Fig. 2)." (line 109 - 110)*

7. Major request
Not true for "collectional activation"
*See main response above.*

8. Major request.

Not true for "collectional activation". They can grow even when 0<S<S_crit if r>r_s.

*This is covered in the study. The corresponding text has been clarified: "In this section, the applied methodology for untangling the contributions of diffusion and collection to the activation of aerosols is introduced. An aerosol becomes activated when it grows beyond its critical radius ($r > r_{crit}$). Moreover, activation requires the particle to be located in a volume of air with a sufficient supersaturation to enable unhindered diffusional growth. Depending on the microphysical process responsible for the final crossing of $r_{crit}$, different supersaturation allow unhindered diffusional growth.*

*Due to the continuous character of diffusional growth, the supersaturation has to be larger than the critical supersaturation in the moment in which the critical radius is exceeded:*

$$S > S_{crit} = S_{eq}(r_{crit}),$$

*where $S_{eq}$ is the equilibrium supersaturation calculated according to Köhler theory (see Eq. (A3)). This condition is automatically fulfilled in the case of diffusional growth due to the constraints of Köhler theory on the equilibrium supersaturation. If the critical radius is exceeded by collection, the radius after collection might be immediately larger than $r_{crit}$ and, hence, the necessary supersaturation is allowed to be smaller to enable unhindered diffusional growth:*

$$S > S_{eq}(r_{ac}),$$

*where $r_{ac} > r_{crit}$ is the wet radius after collection. This criterion is not automatically fulfilled and checked additionally to establish the formal equivalence of both processes, i.e., enabling unhindered diffusional growth after activation. Note that the process of activation, i.e., the entire growth beyond $r_{crit}$, can be driven by diffusional growth or by accumulating liquid water due to collection or by a combination of both." (line 119 - 135)*

9. Major question and suggestion.
In my point of view, the definition of the collectional activation employed here is too complicated and unnatural.
Is it really necessary to include
inact + inact -> inact -> act
inact + act -> inact -> act
inact + act -> act (exclude scavenging)
as collectional activation?
Aren't these very rare events that can be negligible?
Further, I think collectional deactivation should be also interesting.

This is just an idea, but in my opinion, it is better to separate the instantaneous activation/deactivation analysis and history analysis, to clarify the structure of the paper.

It sounds natural to me to define the activation/deactivation categories using only instantaneous information:
* * *
diffusional activation
inact -> act
diffusional deactivation
act -> inact
collectional activation:
inact + inact -> act (only direct one)

collectional deactivation:
inact + act -> inact
act + act -> inact
* * *
For the first step, analyzing the instantaneous activation/deactivation characteristics, should be sufficient.
Then, in the next step, you can carry out history analysis, and indeed it is interesting and important,
However, doing both at once complicate the discussion.
Please consider my proposal.

*Actually, there is only one way to cause a collectional activation in the current study: In the moment a particle grows larger than the critical radius, the integrated collectional mass growth needs to exceed the integrated diffusional mass growth ($\Delta m|_{coll} > \Delta m|_{diff.}$). The various types of interactions have been added to exemplify the naturally occurring microphysical processes that lead to $\Delta m|_{coll} > \Delta m|_{diff.}$ They have been illustrated in Fig. 3 and need to be considered in the interpretation of the results. The only unnatural intervention is the exclusion of scavenging or the collection of drops if $max(r_A, r_B) > r_{crit,C}$. This has been clarified by rewriting the whole paragraph (line 149-166):*

*"To identify a collectional activation, the integrated collectional mass growth $\Delta m|_{coll}$ is compared to the diffusional $\Delta m|_{diff}$ in the moment the particle grows beyond its critical radius. If the former exceeds the latter, $\Delta m|_{coll} > \Delta m|_{diff}$, this activation is considered as collectional. There are various microphysical interactions resulting in $\Delta m|_{coll} > \Delta m|_{diff}$, and its basic types are illustrated in Fig. 3. Note that also a combination or a repetition of these types is possible, i.e., multiple subsequent collections. In a collectional activation of type (i), the water mass growth by collection dominates, i.e., the coalescence of two previously inactivated aerosols A and B results directly or after some diffusional growth in an activated particle C. In a collectional activations of type (ii), the critical radius increases faster than wet radius, i.e., the coalescence of an already activated particle A with another activated or an inactivated particle B results in inactivated particle C, which activates after some diffusional growth. If the resulting particle is directly activated, this process is only considered a collectional activation if the largest wet radius of the two coalescing particles A and B is smaller than the critical radius of the newly produced particle C:*
$$max(r_A, r_B) < r_{crit,C}.$$
*This ensures that the combined water of particles A and B is necessary to activate particle C. If this is not the case, i.e., the water of particle A or B is able to activate particle C on its own, the latter process is considered a regular collection of cloud droplets or as scavenging and neglected in the following analysis. Moreover, the coalescence of two activated particles resulting in a collectional activation is mathematically possible but not found to play a role in the analyzed simulations. Note that only collectional activations of the first type are able to increase the number of activated aerosols, while the second type might have no or a negative impact on the total number of activated aerosols since the coalescence of at least one activated particle results in one activated particle."*

10. Typo
d -> Delta
*Done.*

11. Typo
*Done.*

12. Minor request.
This is ambiguous. Do you mean when it will be activated by diffusion without further coalescence?
*This has been clarified. See answer to comment 9.*

13. Minor request.
Same as above
*This has been clarified. See answer to comment 9.*

14. Typo
*Done.*

15. Typo
*Done.*

16. Minor suggestion.
Do diffusional activations also occur at high altitude? If so, wouldn't it be informative for readers to show also the vertical profile of the diffusional activation?
*Yes, partly because of newly entrained aerosols or due to the kinetically limited activation of aerosols within the central updraft (see, e.g., Slawinska et al. 2012; Hoffmann et al. 2015 as stated in line 200). A vertical profile of the diffusional activation rate has been added (Fig. 5d).*

17. Major question.
This is not trivial. Do you have any clear explanation why this does not happen? Is this just caused by the lack of aerosol particles of this size or is there any other mechanism to inhibit both diffusional and collectional activation?
*For both activation types, the large critical radius inhibits activation for larger aerosols within the typical lifetime of the simulated clouds (about 15 min). The kinetically limited flow of water molecules slows down the diffusional activation at larger radii, e.g., more than 1000 s are necessary for the activation of an aerosol of 1 µm dry radius at 1 % supersaturation (Hoffmann 2016). Similarly, collectional activation is not able to produce the necessary radii in the available time since the droplets might be too small to cause intense collisions. Moreover, the critical radii might be too big for the simulated clouds to sustain them and they might fall out of the cloud before activation (the largest activated aerosol is 200 µm in wet radius, Fig. 7).*

*The questioned sentence has been extended (line 214-215): "Larger aerosols do not activate at all since their critical radius is too large to be exceeded by diffusion or collection."*

18. Minor question.
If red is 0.8 and blue is 0.2, and 100 collectional activations occur, I understand that 80 activated and 20 inactivated aerosols are involved in these

100 collectional activation events. Is this correct?

If so, in Fig.7, red is always larger than blue, but this is puzzling.

At the section starting from L.135, it is declared that the following two processes are considered a collectional activation:

inact + inact -> act

inact + act -> act

It means, the number of activated aerosols involved in collectional activations must be always smaller than the number of inactivated aerosols involved.

However, this is not the case in Fig.7.

Please make this point clear.

Maybe just the legend is opposite? That is, red is inactivated and blue is activated? or maybe you count

inact + inact -> act

as blue and

inact + act -> act

as red?

*The whole figure has been changed to clarify the manuscript. Figure 8 shows the average number of collections necessary for the collectional activation of one aerosol. The number of collected activated or inactivated particles has been neglegted in this figure. All necessary information on how many activated aerosols have been involved in the analyzed collectional activations was contained in the former Fig. 7b (now Fig. 9).*

19. Minor question.

Same question as above. How do you calculate the red and blue line for this case?

*See last comment.*

20. Minor request

It is difficult to follow the meaning of this sentence. In particular the last half. Do you mean "average entrainment height of all particles inside the cloud is the cloud base"?? Please give a clear and detailed explanation.

*Yes. The sentence has been clarified to: "Since multiple collections are necessary for their activation (see Fig. 8), the lower average entrainment height is representative for the average entrainment height of all particles inside the cloud, which is the cloud base through which most particles enter the cloud (e.g., Hoffmann et al. 2015)." (line 263 - 265)*

*References:*

*Boucher, O. (2015). Aerosol–Cloud Interactions. In Atmospheric Aerosols (pp. 193-226). Springer Netherlands.*

*Chuang, P. Y., Charlson, R. J., and Seinfeld, J. H. (1997). Kinetic limitations on droplet formation in clouds. Nature, 390(6660), 594-596.*

*Hoffmann, F., Raasch, S., and Noh, Y. (2015). Entrainment of aerosols and their activation in a shallow cumulus cloud studied with a coupled LCM–LES approach. Atmospheric Research, 156, 43-57.*

*Hoffmann, F. (2016). The Effect of Spurious Cloud Edge Supersaturations in Lagrangian Cloud Models: An Analytical and Numerical Study. Monthly Weather Review, 144(1), 107-118.*

*Khain, A., Ovtchinnikov, M., Pinsky, M., Pokrovsky, A., and Krugliak, H. (2000). Notes on the state-of-the-art numerical modeling of cloud microphysics. Atmospheric Research, 55(3), 159-224.*

*Nenes, A., Ghan, S., Abdul-Razzak, H., Chuang, P. Y., and Seinfeld, J. H. (2001). Kinetic limitations on cloud droplet formation and impact on cloud albedo. Tellus B, 53(2), 133-149.*

*Slawinska, J., Grabowski, W. W., Pawlowska, H., and Morrison, H. (2012). Droplet activation and mixing in large-eddy simulation of a shallow cumulus field. Journal of the Atmospheric Sciences, 69(2), 444-462.*

*Yau, M.K. and Rogers, R.R. (1989). A short course in cloud physics. Elsevier.*

---

## Author Response (AR2)

**Responses to the Referees' Comments**

The authors comments are formatted in bold, italic, blue letters.

**Response to Anonymous Referee #1:**

Thank you for adding the diffusional activation rate to figure 5 and figure 3.

*I am glad that the revised version has been accepted well. Thank you for the second review.*

I have two technical comments regarding figure 5:

Could the collectional and diffusional activation rate plots use the same units on the x axis? It would make it easier to compare the two.

*I have changed the axis of the collectional activation rate to $cm^{-3} s^{-1}$ as used for the diffusional activation rate. See Fig. 5 of the revised question. In that sense, I changed line 211 to: "The comparison of the numerical values of the activation rates in Fig. 5 c and d indicate already …"*

Could you comment why the diffusional activation rate for the more aerosol laden simulations is increasing at the higher levels as compared to the "more clean" conditions?

*There are two points that need to be considered for the interpretation of this rather strange behaviour: First, the maximum supersaturations are decreasing in more polluted conditions, which allows only larger aerosols to activate. On the other hand, larger aerosols have a longer activation time, i.e., they activate at a higher level above the cloud base. I have added this in the lines 200 – 204: "In more aerosol-laden conditions, a larger fraction of diffusional activations occurs at higher levels. In these simulations, only larger aerosols are able to activate by diffusion due to the generally lower supersaturations. These larger aerosols, however, need a longer time to activate. Accordingly, these aerosols are lifted to higher levels by the cloud's updraft until they grow beyond their critical radius for activation with commensurate changes in the profile of the diffusional activation rate."*

**Response to Shin-ichiro Shima:**

Now it becomes clear that a determination criteria appropriate for characterizing the newly introduced concept "collectional activation" is used in this study.

In the original manuscript (L.120), we can read that the determination criteria
$r_{ac} > r_{crit}$ and $S > S_{crit} = S_{eq}(r_{crit})$
is applied also for collectional activation. However, now a rigorous criteria to determine "collectional activation" is specified as follows:
$r_{ac} > r_{crit}$ and $S > S_{eq}(r_{ac})$,
with some detailed explanations.

Opposed to the author's general comment in the response, what I thought was missing in this study is not the 1st scenario, but the 4th scenario, which I explained in the supplement "Possible_collectional_activation_scenario.pdf". There, I tried to explain that even if $S < S_{crit}$, collectional activation can occur if
$r_{ac} > r_s(S)$ (the unstable equilibrium radius for given S)
is satisfied. Indeed this condition is equivalent to
$r_{ac} > r_{crit}$ and $S > S_{eq}(r_{ac})$,
which is now clearly introduced to the manuscript.

*I am thankful that the revised version of the manuscript clarified these aspects. I thank the reviewer for his comments.*

The last thing I request is to correct the 1st sentence of the abstract,
"..., and it occurs as soon as a wetted aerosol grows beyond its critical radius."
This is not true for "collectional activation". As it is now clearly explained by eq.5,
$S > S_{eq}(r_{ac})$ is also required. Maybe just changing as follows would be sufficient.
"..., and it is widely accepted that it occurs as soon as a wetted aerosol grows beyond its critical radius."

*That is a good suggestion. I changed the first sentence of the abstract (lines 1 – 2) as suggested by the reviewer to: "
[revised manuscript text omitted]